# Dog and Guardian Relationships: Application of a Dual-Process Actor–“Partner” Interdependence Model to Predict Regular Walking

**DOI:** 10.3390/bs15050607

**Published:** 2025-05-01

**Authors:** Sarah B. E. Hough, Chris A. Graham, Alfred S. Y. Lee, Ryan E. Rhodes

**Affiliations:** Behavioural Medicine Laboratory, Faculty of Health, University of Victoria, P.O. Box 3010 STN CSC, Victoria, BC V8W 3N4, Canada; christophergraham@uvic.ca (C.A.G.); alfredlee@uvic.ca (A.S.Y.L.); rhodes@uvic.ca (R.E.R.)

**Keywords:** physical activity, actor–partner interdependence model, dog walking, dual-process model

## Abstract

Physical inactivity is a major global health risk, yet many fail to meet activity guidelines. Dog guardianship has been linked to increased physical activity, though the dog–guardian walking relationship remains understudied. This study applied the Actor–Partner Interdependence Model (APIM) to examine how guardians’ and dogs’ dual-process constructs influence walking behaviour. A sample of 127 Canadian dog guardians reported their walking habits, hedonic motivation, and expectations (Time 1) for themselves and their dogs, with follow-up walking behaviour assessed after three weeks (Time 2). Structural equation modelling revealed significant covariation in dog–guardian walking (*r* = 0.38, *p* = 0.03), supporting APIM. Guardians’ hedonic motivation (β = 0.37, *p* = 0.02) and expectations (β = 0.38, *p* = 0.02) predicted both human and dog walking. Findings confirm that guardians are the primary drivers of walking, suggesting interventions targeting guardian motivation and expectations may enhance physical activity in both humans and dogs, benefiting health.

## 1. Introduction

Regular physical activity (PA) plays an important role in general well-being with reduced chances of all-cause mortality, depression, and anxiety in the range of 30–40% ([66]). Physical activity also reduces non-communicable diseases such as diabetes, cardiovascular disease, and certain forms of cancer ([53]). Physical inactivity is currently the fourth-leading risk factor for all-cause mortality globally and is linked with other leading risk factors including elevated blood pressure, increased blood glucose, excess weight, and obesity ([47]; [28]). Despite these considerable benefits, there is a rising prevalence of individuals who are not meeting the suggested physical activity guidelines of 150 min of moderate-intensity physical activity per week ([23]; [59]). Understanding factors associated with physical activity enhances the ability to predict and promote behaviours, and therefore improve health outcomes.

Physical activity (i.e., any bodily movement produced by skeletal muscles that require energy expenditure; [8]) can be organised into different categories according to its purpose. Different modes of physical activity could include household chores (for work), biking to the office (for transport) or walking (for transport or for leisure). Walking is one of the most convenient, widely practiced, and most preferred forms of physical activity as it is accessible, safe, familiar, and inexpensive ([63]; [17]). Walking can also reduce mortality risk by at least 10% with just 30 min of activity most days ([72]). It helps manage menopausal symptoms ([24]), lowers heart rate and sympathetic nerve activity (stress response; [57]), and promotes psychological well-being ([24]). These varied benefits highlight the importance of integrating regular walking into daily routines. Thus, understanding leisure-time walking behaviour is important as it could be a likely means of getting individuals of varied ages and fitness levels to experience physical and psychological benefits, increase the quality and longevity of life, and decrease healthcare spending ([50]; [54]).

One form of walking that may hold utility for health and well-being is dog walking, as many cross-sectional studies have shown dog guardianship is positively associated with meeting physical activity guidelines ([50]; [10]; [31]). This is important, as a substantial number of individuals own dogs; for example, in North America, approximately 31–46% of Canadians own at least one dog and nearly 50% of U.S. households own a dog ([9]; [62]; [50]). Furthermore, a study examining 65 dog guardians in the US found guardians accumulated an average of 23 min of moderate-to-vigorous-intensity PA during daily walks (which would equate to 11 min per week over PA guidelines), while another study found older adult dog guardians spent less time sitting compared to non-guardians (9.4 vs. 10.1 h/day). While promoting dog adoption solely for physical activity is not advisable due to the responsibilities and costs associated with dog guardianship, it is important to recognise there is also considerable variability in dog guardians and walking. For example, some studies have shown that approximately 20% of dog guardians do not have a regular walking routine ([61]), and other studies have claimed that up to 70% of dog guardians do not walk their dog enough to achieve the benefits of an active lifestyle ([10]). As such, understanding dog guardians and walking relationships can be important to help promote walking behaviour in guardians who currently do not walk their dogs often enough.

It is currently known that several factors shape the frequency of dog guardian walking behaviour ([9]; [69]). Guardians with a strong sense of obligation to walk their dogs tend to walk them more frequently, with those owning larger dogs being more consistent in their walking routines ([4]; [48]). Additionally, guardians who believe their dogs enjoy walking are more likely to engage in the activity ([69]). These findings suggest that the bond between a dog and its guardian influences walking behaviours. This association occurs notably with recreational walking, as suggested by [68] ([68]). Dogs could also be a form of social support through companionship, as they co-participate in physical activity (such as walks); this is important to note as relationships and companionship would conceivably influence the cognitions, emotions, and behaviours of one another, at least when it comes to human relationships ([38]).

This idea could help explain why being an overweight dog guardian was found to be the most important factor in the occurrence of obesity in dogs ([60]), or why guardian body mass index is positively correlated with the degree of overweight dogs ([33]; [41]). Obesity in canine companions is concerning as it can shorten their lifespan, decrease quality of life, and increase the risk of comorbidities (i.e., the presence of two or more diseases or medical conditions at once) ([3]). Thus, increasing PA and subsequently improving health can improve the lifespan and quality of life in both humans and dogs.

Despite the correlational evidence between walking frequency with dogs and their guardians, few studies have explored their interaction. One method to explore this dyadic relationship is the Actor–Partner Interdependence Model (APIM). APIM posits that interpersonal interactions “are reciprocal and co-dependent, showing that individuals are fundamentally connected and cannot be considered in isolation” ([73]; [52]). The APIM was first introduced by [13] ([13]) to demonstrate that each party involved in a relationship will influence the cognitions, emotions, and behaviour of one another ([13]). The APIM has been studied in various forms of relationships, such as family relationships (parental–adolescent: [42]), spousal relationships ([14]; [30]), best-friend relationships ([36]), patient–caregiver relationships ([15]), and career relationships (e.g., colleagues; [1]). Some of the studied bidirectional products of these relationships have included work engagement ([1]), work–family interference ([29]), depressive symptoms ([44]), motivation ([6]), burnout ([65]), and physical activity ([7]; [71]; [36]). The diversity in the literature of relationships demonstrates its versatility to be applied to many different outcomes, but although these studies are versatile, they remain limited to human relationships. In the context of dog walking, this model could help explain how both dogs and guardians contribute to walking behaviours.

Just like humans, pet behaviour may result from acquired or innate motivations that affect the relationships they have with their guardians ([34]; [19]). Along with dog-supportive physical environments, the relationship between dogs and their guardians is one of the most effective ways to encourage walking behaviours ([67]). Understanding the behavioural motivation of both guardian and canine likely aligns with a dual-process model, which differentiates between type 1 processes (automatic, habit-driven, and emotionally based behaviours like hedonic motivation) and type 2 processes (deliberate, and expectation-driven behaviours) ([58]).

In terms of walking, while a dog guardian may feel responsible to perform consistent walking behaviours with their pet ([68]), the dog may also develop an expectation (type 2 process) for these walks as they continue to be performed. This may lead the dog to demonstrate walking expectations, which further encourages the guardian to perform such behaviour. In turn, the dog may react accordingly to these expectations. A dog’s expectation could be shown in a variety of demonstrable ways; the dog may run towards their leash, try to go outside, sit by the door, etc. Notably, humans also develop expectations within this dyad ([43]); for example, they may expect their dog to show excitement for a walk throughout the day and use this behaviour as a cue to take them outside. One study examining 3465 prospective dog adopters found 89% of respondents expected increased happiness, 61% expected companionship, 74% expected decreased stress, and 89% of respondents expected increased walking ([43]).

In terms of type 1 processes, if a guardian consistently takes their dog for a walk upon coming home from work, this behaviour is likely to come under strong control of the cue of returning home, and thus become habitual over time by association of walking with that cue ([48]). Hedonic motivation, a learned association between a cue and affective toward a behaviour ([70]), may also form alongside a habit. For example, the feelings of pleasure a dog experiences through going for a walk may lead to future hedonic desire for a walk, especially when put into similar situations as when the initial object of the hedonic motivation has occurred. So, if a dog experiences more frequent walks upon the return of their guardian from work, they may react with hedonic motivation (walk), expressed with behaviours associated with excitement; the same could be true for the guardian’s experience, if they associate these behaviours with taking their dog for a walk at this time. Still, no study has yet to test this interplay between type 1 (hedonic motivation, habit) and type 2 (expectation) processes in dogs and their guardians.

The purpose of this study was therefore to investigate this proposed dual-process model to predict walking across three weeks among dog guardians and their dogs through the framework of an APIM. Although dogs are obviously unable to vocalise their answers, their behaviours and dual-process motivations can be perceived by their guardian as a proxy. With this, we have the following four hypotheses: (1) Dog and guardian walking will share significant covariation, signalling the potential for an APIM approach. (2) In line with the dual-process models, we expect both type 1 (hedonic motivation, habit) and type 2 (expectation) processes to correlate with walking in the independent dog and guardian models (in bivariate correlations and time 1 walking to dual-process models). (3) Time 1 human dual-process constructs will predict time 2 human walking and dog walking behaviours. (4) Commensurate with a dog being an active member of the partnership, we surmise time 1 dog dual-process constructs to predict time 2 dog walking and human walking behaviours. We note that these effects are likely smaller than the same effects from the human guardian, as the dog is not the executive on any walking decisions (i.e., the dog will not walk itself).

## 2. Methods

### 2.1. Study Design

This study featured a three-week prospective, observational design, conducted on Survey Monkey (SurveyMonkey, San Mateo, CA, USA).

### 2.2. Setting

This study was implemented in Canada and had two measured time points. The first survey ran from 20 June to 2 July 2024. The second survey ran from 8 July to 12 August (3 weeks between assessments).

### 2.3. Participants

Participants were invited by a market research firm which has a database of >120,000 Canadian panellists. The panel was representative of the Canadian adult population aged 19 to 65. The panellists were selected to match census data ([22]). Participants who completed the first survey were invited to complete the second survey by the same research firm three weeks later. All participants were living in Canada and able to read and respond to English questions, and self-reported owning a dog. Within the sample of 127 eligible participants, 51.2% were female, 66.1% had a 4-year college degree or higher, 55.1% were full-time employed (14.2% retired), and 43.3% had children living at home. Each participant provided informed consent, and the study was approved by The University of Victoria’s research ethics board.

### 2.4. Variables

An actor–partner model was modified to work for human and dog relationships, with the dog’s guardian answering questions as a proxy for their dog. The following variables were measured during the baseline survey: human walking, dog walking, human habit, dog habit, human hedonic motivation, dog hedonic motivation, human expectations to walk regularly, dog expectations to walk regularly, and participant demographics. The follow-up survey measured human walking and dog walking over their past week.

#### 2.4.1. Human and Dog Walking

The dog guardian walking measure was adapted from the Godin Leisure-Time Exercise Questionnaire ([21]) to align with the Physical Activity Guidelines for Americans ([63]), as well as the Canadian 24-Hour Movement Guidelines ([51]). Participants were asked to recall their average weekly walking exercise for the past week. Participants were provided with descriptions of strenuous, moderate, and mild walking. The questionnaire consisted of three open-ended questions asking weekly frequency of strenuous, moderate, and mild walking, along with three open-ended questions asking their average duration of strenuous, moderate, and mild walking in minutes in alignment with public health recommendations. Only responses to moderate–vigorous intensities were included in the analysis. Commensurate with Canadian PA guidelines and similar to [32] ([32]), the criteria for bout duration were adjusted to 20 min or more for vigorous-intensity walking and 30 min or more for moderate-intensity walking, aligning with Physical Activity Guidelines for Americans ([63]). These bouts were then aggregated to calculate a composite total frequency of moderate-to-vigorous (MV) walking bouts for participants ([32]).

Dog walking was measured using an adapted version of the Godin Leisure-Time Exercise Questionnaire ([20]) modified from previous research ([4]; [32]; [46]) to align with the Physical Activity Guidelines for Americans ([63]). Participants were asked to recall their average weekly dog walking over the past week and were provided with descriptions of strenuous, moderate, and mild walking. The questionnaire consisted of three open-ended questions to measure the dog’s weekly frequency of strenuous, moderate, and mild walking, along with three open-ended questions measuring the dog’s average duration of strenuous, moderate, and mild walking in minutes. These questions were used to calculate the total time the dogs spent doing strenuous, moderate, and mild walking for the week. Only responses to moderate–vigorous intensities were included in the analysis. The criteria for bout duration were adjusted to 20 min or more for vigorous-intensity walking and 30 min or more for moderate-intensity walking, aligning with Physical Activity Guidelines for Americans ([63]). These bouts were then aggregated to calculate a composite total frequency of moderate-to-vigorous (MV) walking bouts for the dog by multiplying the frequency and duration of qualifying bouts and summing across intensities ([32]).

#### 2.4.2. Habit

The guardian’s walking habit was assessed using four questions from the automaticity subscale ([18]) of the self-reported habit index ([64]). An example item was “regular walking is something that I am ready to do automatically”. Each of these questions was rated on a 7-point Likert scale by the participant, from (1) strongly disagree to (7) strongly agree. In the present study, the Cronbach’s alpha coefficient of this scale was 0.96. The dog’s walking habit was assessed using the same four questions from [18] ([18]) but modified to consider the guardian’s dog. Example items include “regular walking is something that my dog is ready to do automatically”, and “regular walking is something that my dog is ready to do without having to be reminded”. Each of these questions was rated on a 7-point Likert scale by the dog guardian proxying for their dog. Internal consistencies of the scales’ scores (α = 0.94) were excellent.

#### 2.4.3. Hedonic Motivation

Guardian hedonic motivation was assessed using one question from [16] ([16]), as recommended by [70] ([70]), with a valence ranging from (1) “walking is something I dread” to (7) “walking is something I look forward to”. Dog hedonic motivation was assessed using the same single item question modified from (1) “walking is something my dog dreads” to (7) “walking is something my dog looks forward to”.

#### 2.4.4. Expectation to Walk

Guardian walking expectation was measured with one question modified from [49] ([49]): “I expect to go on regular *walks*”. The question was scored on a 7-point Likert scale, from (1) *never* to (7) *always*. Dog expectation was measured with the same question modified to proxy for the guardian’s opinion of their dog’s expectations about regular walking: “My dog expects to go on regular walks”. The question was also scored on a 7-point scale from (1) never to (7) always.

#### 2.4.5. Demographics

Dog guardian demographics were measured at baseline, assessing participants’ age, income, gender, ethnicity, employment, level of education, household status, height, weight, and health status.

### 2.5. Statistical Methods

For the preliminary analyses, we examined mean, standard deviations, and correlations of the study variables. Independent sample *t*-tests were conducted to identify any significant differences in demographic characteristics (i.e., gender, age, education level, income, employment status, and health profile) and study variables (i.e., guardian’s and dog’s habit, hedonic motivation, expectations, and walking behaviours) between the participants who completed the post-test survey and those who did not. Preliminary analyses were conducted using SPSS v26 ([26]).

To test the hypotheses, we used structural equation modelling (SEM) to examine the proposed APIM. First, we tested a model without controlling for baseline walking behaviour (Figure 1). This model included guardians’ habit, hedonic motivation, and expectations, as well as dogs’ habits, hedonic motivation, and expectations at baseline, to predict follow-up walking behaviour. Then, we tested an alternative model that controlled for baseline walking behaviour (Figure 2). An a priori power analysis was conducted. Using the RMSEA approach for structural equation modelling ([37]) implemented, we contrasted a exact-fit model (RMSEA = 0.00) with a not-close model (RMSEA = 0.08) and set α = 0.05 (two-tailed) and desired power (1 − *β*) = 0.80. With 48 degrees of freedom (*df*) in the hypothesised APIM and 60 *df* in the baseline-controlled model, the critical sample sizes were *N* = 97 and *N* = 85, respectively. Analyses were conducted in Mplus version 7.2 ([40]). Model fit was assessed using conventional fit indices, with the following thresholds indicating acceptable fit: Comparative Fit Index (CFI) and Tucker–Lewis Index (TLI) values ≥0.90, and root mean square error of approximation (RMSEA) and standardised root mean square residual (SRMR) values <0.08 ([25]). Missing data were handled using the full information maximum likelihood (FIML) method, which computes case-wise likelihood using observed variables ([40]).

## 3. Results

### 3.1. Descriptive Data

A total of 127 eligible respondents (*M*_age_ = 47.03 ± 11.68, range = 22 to 65; female = 51.2%) agreed to participate in the present study and completed the first online survey. Table 1 summarises their demographic information. Of these, 43 did not complete the post-test survey, resulting in a retention rate of 66%. Missing data were analysed using Little’s missing completely at random (MCAR) test ([35]), which indicated no systematic pattern in the missing data (i.e., χ^2^ = 67.76, *df* = 71, *p* = 0.59). Regarding the dropout analysis, the results of the independent sample *t*-tests revealed no significant differences in baseline demographic information or study variables between participants who dropped out and those who remained (*t*(1, 125) = −1.15 to 1.98, *p*s = 0.05 to 0.90, except for guardian’s expectations). Participants who dropped out had significantly higher expectations about regular walking at baseline than those who remained (*t*(1, 125) = 2.05, *p* = 0.04). Descriptive statistics, including mean, standard deviations, skewness, kurtosis, and correlations of study variables are presented in Table 2. The four walking variables were identified as non-normally distributed (skewness ≥ 1.93 and kurtosis ≥ 3.52). As a result, these variables were square root transformed ([2]) to meet normality assumptions (skewness = 0.52 to 1.07 and kurtosis = −0.61 to 2.78).

### 3.2. APIM Models

The proposed APIM (Figure 1) demonstrated a good fit to the data: χ^2^ = 75.22, *df* = 48, CFI = 0.97, TLI = 0.95, RMSEA = 0.07 (90% CI = 0.04–0.10), SRMR = 0.03. Guardians’ hedonic motivation significantly predicted their own follow-up walking behaviour (*β* = 0.37, *p* = 0.02) and marginally predicted their dogs’ follow-up walking behaviour (*β* = 0.20, *p* = 0.06). Dogs’ follow-up walking was also significantly predicted by guardians’ baseline expectations (*β* = 0.38, *p* = 0.02). However, dogs’ habit, hedonic motivation, and expectations did not significantly predict either their own (*β*s = −0.03 to 0.10, *p*s = 0.46 to 0.83) or their guardians’ follow-up walking behaviour (*β*s = −0.25 to 0.15, *p*s = 0.10 to 0.40). A significant correlation between guardians’ and dogs’ walking behaviour at follow-up was observed (*r* = 0.38, *p* < 0.03).

When baseline walking behaviours were controlled in the model, the proposed APIM (Figure 2) demonstrated an acceptable fit to the data: χ^2^ = 106.43, *df* = 60, CFI = 0.96, TLI = 0.92, RMSEA = 0.08 (90% CI = 0.05–0.10), SRMR = 0.03. At baseline, guardians’ walking behaviour was significantly associated with their habit, hedonic motivation, and expectations (*β*s = 0.34 to 0.51, *p* < 0.01). Dogs’ walking behaviour at baseline was also significantly linked with their habit (*β* = 0.24, *p* < 0.01) and expectations (*β* = 0.42, *p* < 0.01) but not with hedonic motivation (*β* = 0.16, *p* = 0.17). No significant cross-associations were found between guardians’ walking behaviour and dogs’ habit, hedonic motivation, or expectations at baseline (*β*s = −0.03 to 0.10, *p*s = 0.31 to 0.87), nor between dogs’ walking behaviour and guardians’ habits, hedonic motivation, or expectations (*β*s = −0.04 to 0.13, *p*s = 0.14 to 0.70). For follow-up walking behaviour, guardians’ hedonic motivation (*β* = 0.31, *p* = 0.05) and their baseline walking behaviour (*β* = 0.40, *p* = 0.04) were the only marginally significant and significant predictors of their follow-up walking behaviour. Meanwhile, guardian’s expectations (*β* = 0.23, *p* = 0.07) and dogs’ baseline walking behaviour (*β* = 0.37, *p* = 0.04) had marginally significant and significant effects, respectively, on dogs’ follow-up walking behaviour. In contrast to the model without controlling for baseline walking behaviour, guardians’ hedonic motivation did not have significant effects on their dogs’ follow-up walking behaviour (*β* = 0.12, *p* = 0.18). Dogs’ habit, hedonic motivation, and expectations did not significantly predict their own follow-up walking behaviour (*β*s = −0.01 to 0.11, *p*s = 0.42 to 0.92) or that of their guardians’ (*β*s = −0.25 to 0.13, *p*s = 0.14 to 0.46). Significant and marginally significant correlations were observed between guardians’ and dogs’ walking behaviour at baseline (*r* = 0.63, *p* < 0.01) and follow-up (*r* = 0.32, *p* = 0.08).

## 4. Discussion

The purpose of this study was to examine the dynamic interplay between dog and guardian walking behaviours, using a modified APIM and dual-process theoretical framework. The first hypothesis was that dog and guardian walking would share significant covariation, signalling the potential for an APIM approach (i.e., do active guardians typically have more active dogs?). As expected, this hypothesis was confirmed. Specifically, dog and human walking showed a medium-sized correlation (r = 0.38) which supports the application of an APIM ([11]). Correlational analysis also showed that dog walking was significantly associated with guardian walking at both baseline (r = 0.63) and follow-up (r = 0.32). These outcomes align with and directly test prior research, which has generally shown that dog guardians walk more than non-guardians, presumably due to walking their dogs. While this evidence is correlational, it supports the mutual influence within the dyad. This finding is also supported by previous research; for example, a review by [56] ([56]) found that guardians who regularly walk their dogs are 2.5 times more likely to meet the moderate-intensity physical activity guidelines than their non-dog-owning counterparts. This effect also directly substantiates the presumed benefit of canine-facilitated interventions. A review by [45] ([45]) found 82% of studies favoured canine-facilitated interventions for increasing physical activity. In other words, it is possible that when the health of someone else (such as our pet, our companion) is the focus of the intervention, we may be more inclined to follow suit (i.e., increase walking behaviours). Notably, this highlights the potential for interventions to leverage the dog–guardian relationship in improving physical activity frequency ([45]). The result of this hypothesis additionally signifies a dyadic relationship, which supports the use of a dual-process model in the second hypothesis.

Building on this result, it was hypothesised that both type 1 (hedonic motivation, habit) and type 2 processes (expectation) would be associated with walking behaviour in both dogs and guardians. In line with the assumptions of dual-process models ([58]), our results supported this hypothesis, showing that both type 1 and type 2 processes correlated with walking and were independent predictors of time 1 walking in the dog and guardian models. Specifically, hedonic motivation was a significant predictor of guardian walking, demonstrating its importance for sustained activity. Expectation also played a role, particularly in predicting dog walking. The guardian’s hedonic motivation significantly predicted both their own and their dog’s walking behaviour, though the effect on dog walking was weaker. This ultimately tells us that the guardian’s situational motivation in the form of hedonic motivation (want vs. dread) is key to both the guardian’s walking and the dog’s walking, although some walking also appears driven from guardian expectations. In contrast, the dog’s dual-process constructs did not significantly predict their own walking, reinforcing the idea that human behaviour serves as the primary driver of dog activity.

These results emphasise the importance of a dual-process model in comparison to past research which predominantly focused on social cognitive (type 2) explanations of motivation ([67]) in understanding walking behaviours for dogs and their guardians. Dual-process models may be more suited to evaluate this behaviour compared to other models for a variety of reasons. While social cognitive models tend to assume behaviour is consciously planned, studies have shown that dog walking often persists due to habit, even in the absence of intention, even such that habit strength is a stronger predictor of walking than intention is ([48]). Human habit formation may create expectations in dogs, which could reinforce consistent walking behaviour (for example, if a walking habit is created first thing in the morning, then the dog may develop an expectation to walk once their guardian has woken up, which reinforces the guardian to walk). This interplay of automatic processes and the dog’s expectations should therefore target not only motivation but also habit formation to ensure long-term adherence.

Interestingly, while guardian expectations significantly predicted dog walking, they did not predict guardian walking at follow-up. This could suggest that expectations represent consistent routines that dogs respond to, rather than motivation of the guardian. If the dog is highly receptive to predictable behavioural patterns, a guardian’s expectation to walk may be demonstrated in ways which signal walking to the dog (putting on shoes, grabbing a leash), thus influencing the dog’s walking. On the contrary, human walking may be more dependent on fluctuating factors such as mood, time availability, or reactions to various weather patterns. (Have you ever noticed more dog walkers out when there is a break in bad weather/showers or when it is sunny versus when it is raining outside?)

The main analyses, however, focused on the predictive pathways from time 1 constructs to time 2 walking behaviours. Hypothesis 3 focused on time 1 human dual-process constructs predicting time 2 human and dog walking behaviours. This was clearly supported. Specifically, guardian’s hedonic motivation and habit had a significant, medium-sized effect on walking. This demonstrates the pivotal role of hedonic motivation driving human walking within the dyad, as guardians with strong hedonic motivation and well-established habits were more likely to maintain consistent walking behaviours. Hedonic motivation has been a predictor of physical activity in past research ([5]). It is possible that hedonic motivation could be increased by the environment surrounded by the walkers (e.g., land use, land attractiveness and safety) acting as a cue to the associated affect of walking, consistent with the findings of [39] ([39]). There is, however, a scarcity of research relating to hedonic motivation, particularly when compared to other forms of affect-based motivation. Associative conditioning could alternatively be used in light of this, as people with positive automatic affective evaluations of physical activity are generally more active ([12]). Since these people are more active, perhaps exploring how associative conditioning strengthens hedonic motivation is more attainable.

In contrast to the other hypotheses, there was no support for our fourth hypothesis, that time 1 dog dual-process constructs would predict time 2 dog walking and human walking behaviours. While dogs do contribute to the dyad in bivariate analyses, the independent association of motivation is not as strong as that of their guardians. The lack of effect was unanticipated, especially given the previous literature showing dog energy levels have been found to explain 30% of walking behaviour ([32]), and that motivation dogs provide has also increased walking behaviours ([67]). It could be that some guardians do not necessarily enjoy walking with their dog (perhaps it induces stress, due to behavioural issues of their own dog or fear of the behavioural issues of other dogs; perhaps the dog stops to sniff too often, which does not align with the guardian’s walking goals, subsequently causing the guardian to leave the dog at home when they go for their own walks). It is also possible that the finding does not hold after controlling for human motivation, thus showing the importance of APIM modelling. Finally, expectations played a more important role in shaping dog walking patterns (r = 0.50, *p* < 0.1), demonstrating their sensitivity to consistent cues given by their guardians. One potential explanation could be that dogs experience motivation to walk often throughout the day (if it is a highly energetic dog, they may be motivated all day for such an activity), whereas they only demonstrate behaviours for this motivation when they have an expectation.

Despite the refuted fourth hypothesis, this study has nonetheless provided theoretical and practical insight into the differential roles of human and dog constructs in their relation to physical activity and walking behaviours. The results suggest that interventions aimed at promoting physical activity among dog guardians should focus on enhancing hedonic motivation and habit and establishing consistent walking cues for both the guardian and the dog. The human will, however, remain the main focus for both themselves and their pet. Interventions should be targeted with a focus mainly on the motivation of the guardian. Research aiming to increase physical activity motivation has often used methods like cognitive–behavioural therapy in this regard ([55]; [27]).

### Limitations

Limitations of this study include the self-reported nature of behaviour, and therefore, it could introduce biased reports of physical activity. The use of objective tracking materials such as Smartwatches (and fitbark for dogs) could therefore be used for more accuracy. Also, although our study’s attrition rate was modest (i.e., 34%), the MCAR test and dropout analysis suggest that the missing data were largely random. Nonetheless, caution is advised when interpreting the results, as attrition can still reduce the generalizability and precision of the findings. Another limitation is that guardians in this study completed the survey as a proxy for their dogs. The guardian’s perspective of their dog may not fully capture the dog’s true motivations and/or behaviours. Direct observation from a third party and/or animal expert (such as a veterinarian or animal behaviour analyst) could aid in providing further insight on these matters. The presence of a third-party observer could, however, alter the behaviour of the dog, which could further skew data collection. Finally, the sample used in this research did not provide demographics on their dogs, so it is impossible to determine whether dog size, age, breed, health status, etc., had an impact on the results. While dog health and age were not a strict inclusion criterion, participants were asked to report only moderate-to-vigorous walking behaviour, which implicitly excludes dogs that are not regularly walked due to physical limitations.

Future research in this area could therefore control for these affects and include more detailed canine demographic and health profiles to enhance precision and generalizability of the dual-process APIM. It is important to note that by only including moderate-to-vigorous walking, this measurement may have excluded dogs with limited mobility or other health conditions, thus partially controlling for variability in dog walking ability. Health status for guardians could additionally be recorded for enhanced precision, and a full assessment of the dog demographics would better assist future research. Similarly, dogs may have been walked by other caretakers. These nuances were not captured and should be considered in future work that aims to differentiate shared versus separate activity patterns within pet–owner dyads. Future research could also explore the application of APIM in other contexts of human–animal interactions, such as in multi-pet households, family dog-owning households versus independent dog-owning households, etc. Longitudinal studies evaluating the change in physical activity pre-guardianship to post-guardianship is another potential area where this could be explored.

## 5. Conclusions

In summary, this study affirms the interdependence of walking behaviours between dogs and their guardians. The guardian’s hedonic motivation and habit significantly predicted both their own and their dog’s walking behaviour, though the effect on the dog was weaker. In contrast, the dog’s dual-process constructs did not predict their own walking after controlling for human dual-process constructs, which reinforces the idea that humans are the primary drivers of this activity. The findings of this study display the value of dog–guardian relationships in promoting physical activity interventions which can, in turn, contribute to improving global human and canine health, especially given the considerable number of homes that own a dog.

## Figures and Tables

**Figure 1 behavsci-15-00607-f001:**
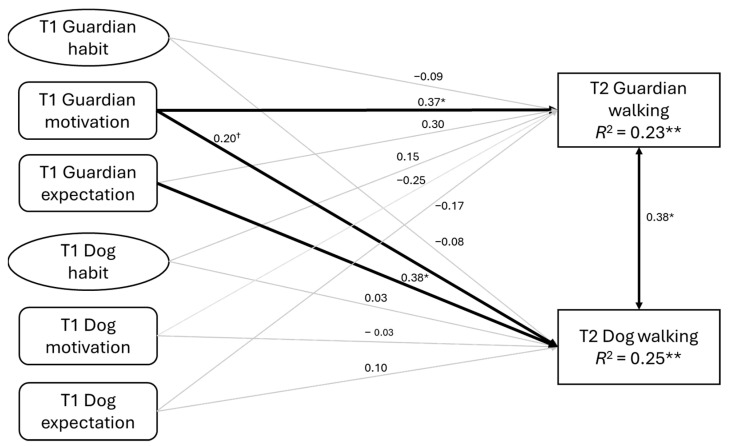
The proposed Actor–Partner Interdependence Model without controlling for walking behaviour at baseline. Note. All the paths were standardised parameter estimates. T1 = Baseline; T2 = Follow-up. † *p* < 0.10, * *p* < 0.05, ** *p* < 0.01.

**Figure 2 behavsci-15-00607-f002:**
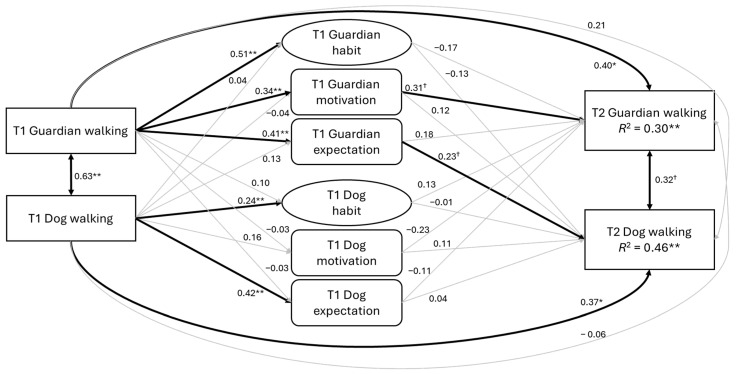
The proposed Actor–Partner Interdependence Model controlling for walking behaviour at baseline. Note. All the paths were standardised parameter estimates. T1 = Baseline; T2 = Follow-up. † *p* < 0.10, * *p* < 0.05, ** *p* < 0.01.

**Table 1 behavsci-15-00607-t001:** Participants’ demographic information.

Dog Guardian Demographic Profile	
Age in years (*SD*)	47.03 (11.68)
% Female	51.2
% Caucasian	77.2
% 4-year college and above	66.1
% Income $100 k and above	41.0
% Full-time employed	55.1
% Retired	14.2
% Children living at home	43.3
Health Profile	
Body mass index (*SD*)	16.48 (4.58)
% Smoker	18.6
% Angina	0.0
% Heart attack	0.0
% Stroke	1.6
% Type1 diabetes	1.6
% Type2 diabetes	4.7
% Gestational diabetes	1.6
% Cancer	3.1
% High blood pressure	11.0
% High blood cholesterol	12.6
Self-Reported Health	
% Poor	5.5
% Fair	13.4
% Good	44.1
% Very good	23.6
% Excellent	6.3

**Table 2 behavsci-15-00607-t002:** Zero-order correlations, and descriptive statistic of the study variables (*N* = 127).

Variables	1	2	3	4	5	6	7	8	9	10
1. T1 Guardian habits	1									
2. T1 Guardian motivation	0.65 **	1								
3. T1 Guardian expectation	0.80 **	0.69 **	1							
4. T1 Dog habits	0.43 **	0.42 **	0.42 **	1						
5. T1 Dog motivation	0.21 *	0.43 **	0.25 **	0.56 **	1					
6. T1 Dog expectation	0.44 **	0.40 **	0.50 **	0.67 **	0.50 **	1				
7. T1 Guardian MV walking	0.36 **	0.11	0.28 **	0.17	−0.07	0.11	1			
8. T1 Dog MV walking	0.32 **	0.11	0.31 **	0.25 **	0.02	0.30 **	0.73 **	1		
9. T2 Guardian MV walking	0.12	0.35 **	0.10	0.02	−0.10	0.09	0.33 **	0.07	1	
10. T2 Dog MV walking	0.36 **	0.40 **	0.47 **	0.27 *	0.24 *	0.31 **	0.44 **	0.43 **	0.47 **	1
Mean	4.18	4.89	5.06	5.50	5.73	5.52	3.65	3.50	2.80	2.34
SD	1.91	1.75	1.79	1.60	1.69	1.92	6.65	4.85	5.10	3.65
Skewness	−0.48	0.55	−0.85	−1.44	1.22	−1.29	6.50	2.71	4.08	1.93
Kurtosis	−0.91	−0.46	0.28	1.58	0.43	0.57	57.67	13.22	21.71	3.52

Notes. MV = moderate-to-vigorous; T1 = baseline; T2 = Follow-up. * *p* < 0.05, ** *p* < 0.01.

## Data Availability

The data presented in this study are available on request from the corresponding author due to ethical reasons.

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
