# Peer review of "Dog and Guardian Relationships: Application of a Dual-Process Actor–“Partner” Interdependence Model to Predict Regular Walking"

_behavsci, 2025, doi:10.3390/bs15050607_

Round 1

Reviewer 1 Report

Comments and Suggestions for Authors

Thank you for exploring this important topic as it relates to both human and animal health. 

Line 64- Please ensure PA is added previously in the manuscript to denote Physical Activity.

Line 158- Add City and State of business Survey Monkey

Line 164-All participants were living in Canada and able to read/respond English questions, who self-reported owning a dog. The study would be strengthened by adding inclusion criteria about dog owners that have a dog that is regularly walked?  I may be missing something but your model is dependent on 2 time points and understanding the dogs behaviour and how the dogs behaviour contributes to the dog owners behaviour. Without the demographics and health status of the dogs in the study, I think your conclusion may be flawed. Owning a geriatric dog with extensive health issues vs a young active dog will impact the dogs signaling and ability to want to go on walks. Unless you feel the only selecting Moderate activity reported controls for that in your model. Please discuss in more depth why you feel that the results are still accurate with out this background information.

Line 379- Hedonic motivation should thus be a key in developing long-term engagement of dog walking behaviours. Please remove thus in this sentence.

Line 428- Please add to the dog demographics, the dog health status was not collected as was done for the dog owners.  Age and Health conditions which limit the dog's ability to walk moderately such as untreated OA are not accounted for this in this study. It would be beneficial in future studies to draw conclusions about dogs motivation and behaviour. 

A Gap in the background in references to the dogs' wellbeing from exercise in the Introduction. Adding a few references on why exercise is also good to prevent obesity in dogs would also be beneficial to this article. The most focus is on the dog owner but obesity and inactivity are also important to canine health. 

Author Response

Reviewer 1:

Thank you for exploring this important topic as it relates to both human and animal health. We thank the reviewer for this positive disposition about our paper and have addressed the comments below.

 Line 64- Please ensure PA is added previously in the manuscript to denote Physical Activity. As requested, we now indicate that PA denotes physical activity on Line 28.

Line 158- Add City and State of business Survey Monkey. As requested, we now indicate the state of business Survey Monkey (Line 168).

Line 164-All participants were living in Canada and able to read/respond English questions, who self-reported owning a dog. The study would be strengthened by adding inclusion criteria about dog owners that have a dog that is regularly walked?  I may be missing something but your model is dependent on 2 time points and understanding the dogs behaviour and how the dogs behaviour contributes to the dog owners behaviour. Without the demographics and health status of the dogs in the study, I think your conclusion may be flawed. Owning a geriatric dog with extensive health issues vs a young active dog will impact the dogs signaling and ability to want to go on walks. Unless you feel the only selecting Moderate activity reported controls for that in your model. Please discuss in more depth why you feel that the results are still accurate with out this background information. Thank you for this thoughtful feedback. We agree that characteristics such as age, breed, and health status of the dog can influence both the dog’s behaviour and walking patterns, which in turn could impact the owner’s walking behaviours. In this study, we focused on moderate-to-vigorous walking to reflect more health-relevant activity and help filter out cases where the dog is unable/unwilling to walk due to severe limitations. We recognize, however, that this does not fully account for variations in dog demographics, and we have therefore also added this point to the limitations section (Lines 471-473; 465-467).

Line 379- Hedonic motivation should thus be a key in developing long-term engagement of dog walking behaviours. Please remove thus in this sentence. As requested, this sentence has been removed from the manuscript.

Line 428- Please add to the dog demographics, the dog health status was not collected as was done for the dog owners.  Age and Health conditions which limit the dog's ability to walk moderately such as untreated OA are not accounted for this in this study. It would be beneficial in future studies to draw conclusions about dogs motivation and behaviour. As requested, this has been added to the limitations of the study (Lines 464-478).

A Gap in the background in references to the dogs' wellbeing from exercise in the Introduction. Adding a few references on why exercise is also good to prevent obesity in dogs would also be beneficial to this article. The most focus is on the dog owner but obesity and inactivity are also important to canine health. We agree that including more background in dogs’ wellbeing would add relevant substance to the introduction. Some references have been added to the introduction in congruence with this suggestion (Lines 87-94).

Reviewer 2 Report

Comments and Suggestions for Authors

General Comments: I had the pleasure of reviewing the study “Dog and Owner Relationships: Application of a Dual Process Actor-“Partner” Interdependence Model to Predict Regular Walking”. Overall, the study was well-written, if somewhat technically dense. The manuscript was reasonable to read and the methods seemed logical. The results, although different from expected by the study team, were not earth-shattering. This could be my perception because the discussion could use a little more storytelling about why these results matter and how to apply them.

One major comment is that the animal behavior field is moving away from the word “owner” and toward the use of “caretaker” or “guardian” or “family”. Think about switching the word “owner” out for something else.

Introduction

  • Line 66: I really liked this thought: “While promoting dog adoption solely 66 for physical activity is not advisable due to the responsibilities and costs associated with dog ownership”
  • Line 120”: Please remove the word “deliberate”. We try to avoid saying that any particular behavior is “deliberate” or not, regardless of animal or person. Every operant behavior that occurs is done for a reason, and therefore all operant behaviors are deliberate. I wonder if what you really mean is something like “demonstrable” or perhaps you mean to say that the dog engages in behaviors which function to produce walks?
  • Line 128: I don’t know what “automatic” behavior is. In behavior analysis, we would describe this as “this behavior is likely to come under strong control of the cue of returning home.” Could that work instead?
  • Line 136: I’m not sure that pleasure is an emotion, and I have no idea what it looks like. Perhaps just label this as “expressed with behaviors associated with excitement.”
  • Line 153: I really like this notation that the dog is not truly the executive.

Methods

  • I’m not following what the purpose of doing surveys at two timepoints was. I could see this being useful if you were measuring behavior in owners who had just acquired a dog and were comparing what they THOUGHT would happen in the future versus how much they actually walked/wanted to walk. But these are all owners who already owned dogs and already have behavior patterns of walking/not walking right? So is the purpose of the second survey to assess the consistency of the responses across time? Perhaps describe in more detail what the purpose of two timepoints of surveys was.
  • Did you control for whether the owners were walking WITH the dog? Because I would wonder whether many owners do walk but don’t take their dogs with them (whether because of medical issues or behavioral issues).

Results

No comments

Discussion

  • Overall, I would ask the authors to make the discussion more of a story and less technical. Allow for readers with less statistics- or modeling-heavy backgrounds to fully understand what your results suggest and how they can be applied. Therefore, I suggest simplifying the discussion and reformatting to tell more of a story, including how to apply these results.
  • It seems odd that owner expectations were associated with T2 dog walking, but not T2 owner walking. Can you explain this?
  • Can you provide simple data which indicate and describe whether dog walking predicted owner walking? I believe this is .63 under T1 and .32 under T2, but I think simple outcome deserves some discussion – that if dogs are walking, then their owners are more likely to do so as well. Again, this is important as not all dogs are able to be walked. I think this likely at least partially explains your sentence in Line 353 (“Owner’s hedonic motivation significantly predicted both their own and their dog’s walking behavior, though the effect on dog walking was weaker.”
  • I know I keep stating this, but in paragraph 387-398, you describe how the dog’s constructs don’t seem to predict the walking behavior. Again, I think this could be due to medical/behavioral issues. A dog may be super excited about walking and yet still be unable to do it with their owner, or perhaps the owner does not find walking WITH the dog to be enjoyable and thus walks alone. For example, many dogs are not well trained on a leash and therefore pull and stop quite a bit, which can be frustrating to owners who just want to keep walking at a particular pace for a long period of time.

Author Response

Reviewer 2:

General Comments: I had the pleasure of reviewing the study “Dog and Owner Relationships: Application of a Dual Process Actor-“Partner” Interdependence Model to Predict Regular Walking”. Overall, the study was well-written, if somewhat technically dense. The manuscript was reasonable to read and the methods seemed logical. The results, although different from expected by the study team, were not earth-shattering. This could be my perception because the discussion could use a little more storytelling about why these results matter and how to apply them. We would like to thank the reviewer for this positive feedback. We have adjusted the discussion to have more of a store-telling feel to it, and hope this adds more substance to the paper. We have additionally addressed the comments below and thank you for the thoughtful insights.

One major comment is that the animal behavior field is moving away from the word “owner” and toward the use of “caretaker” or “guardian” or “family”. Think about switching the word “owner” out for something else. As requested, we have replaced the word “owner” to “guardian” throughout the entirety of the paper.

Introduction

  • Line 66: I really liked this thought: “While promoting dog adoption solely 66 for physical activity is not advisable due to the responsibilities and costs associated with dog ownership”. We would like to thank you for this positive comment!
  • Line 120”: Please remove the word “deliberate”. We try to avoid saying that any particular behavior is “deliberate” or not, regardless of animal or person. Every operant behavior that occurs is done for a reason, and therefore all operant behaviors are deliberate. I wonder if what you really mean is something like “demonstrable” or perhaps you mean to say that the dog engages in behaviors which function to produce walks? As requested, the word “deliberate” has been replaced with “demonstrable” (Line 129).
  • Line 128: I don’t know what “automatic” behavior is. In behavior analysis, we would describe this as “this behavior is likely to come under strong control of the cue of returning home.” Could that work instead? This change has been made to the manuscript as requested (Lines 137-138).
  • Line 136: I’m not sure that pleasure is an emotion, and I have no idea what it looks like. Perhaps just label this as “expressed with behaviors associated with excitement.” This change has been made to the manuscript as requested (Line 145).
  • Line 153: I really like this notation that the dog is not truly the executive. We thank you for this positive feedback.

 Methods

  • I’m not following what the purpose of doing surveys at two timepoints was. I could see this being useful if you were measuring behavior in owners who had just acquired a dog and were comparing what they THOUGHT would happen in the future versus how much they actually walked/wanted to walk. But these are all owners who already owned dogs and already have behavior patterns of walking/not walking right? So is the purpose of the second survey to assess the consistency of the responses across time? Perhaps describe in more detail what the purpose of two timepoints of surveys was. We thank you for this valuable thought. We employed a two-timepoint design to capture prospective relationships among both owner and dog variables. At Time 1, participants reported baseline walking habits and motivational constructs (e.g., expectations, hedonic motivation). Three weeks later (Time 2), we reassessed walking behavior. This interval allowed us to observe whether initial habits and motivational constructs at Time 1 would predict subsequent walking (both for the owner and the dog) at Time 2. The design thus helps establish a temporal (rather than merely cross-sectional) link between baseline cognitions and follow-up behaviors.
  • Did you control for whether the owners were walking WITH the dog? Because I would wonder whether many owners do walk but don’t take their dogs with them (whether because of medical issues or behavioral issues). We wanted these to be independent, as per the actor-partner independence model. This way we can explore the covariance of walking between the two targets (dog and owner). We recognize that the dog may not walk independent of the owner, but it is possible with a mixed schedule of different caretakers. We have highlighted these points in the discussion (Lines 425-429; 471-478).

 Discussion

  • Overall, I would ask the authors to make the discussion more of a story and less technical. Allow for readers with less statistics- or modeling-heavy backgrounds to fully understand what your results suggest and how they can be applied. Therefore, I suggest simplifying the discussion and reformatting to tell more of a story, including how to apply these results.Thank you for this suggestion! We have made an effort implement more real-life examples in to the discussion in order to implement more of a relatable, “story-telling” feel to the discussion. We hope this makes a difference to the overall feel of the paper.
  • It seems odd that owner expectations were associated with T2 dog walking, but not T2 owner walking. Can you explain this? Thank you for the valuable observation – we agree that it appears somewhat counterintuitive that owner expectations predicted follow-up dog walking but not their own walking. We would like to put forward the idea that in this context, expectation may reflect cues that the dog responds to, rather than the motivation for the owner’s own activity.  While these cues remain consistent, owner walking may be influenced by a broader range of factors. We have expanded on this in lines 398-405.
  • Can you provide simple data which indicate and describe whether dog walking predicted owner walking? I believe this is .63 under T1 and .32 under T2, but I think simple outcome deserves some discussion – that if dogs are walking, then their owners are more likely to do so as well. Again, this is important as not all dogs are able to be walked. I think this likely at least partially explains your sentence in Line 353 (“Owner’s hedonic motivation significantly predicted both their own and their dog’s walking behavior, though the effect on dog walking was weaker.” The data has been provided in the discussion as requested. We agree that the dog’s ability to walk is an important factor to be considered here (Lines 346-348).
  • I know I keep stating this, but in paragraph 387-398, you describe how the dog’s constructs don’t seem to predict the walking behavior. Again, I think this could be due to medical/behavioral issues. A dog may be super excited about walking and yet still be unable to do it with their owner, or perhaps the owner does not find walking WITH the dog to be enjoyable and thus walks alone. For example, many dogs are not well trained on a leash and therefore pull and stop quite a bit, which can be frustrating to owners who just want to keep walking at a particular pace for a long period of time. Thank you for this feedback, and we apologize for not touching on this important point in the first submission. Although the inclusion of exclusively moderate-to-vigorous physical activity may have avoided dogs with health issues, we had not mentioned any factors such as behavioural issues or enjoyment in walking without the dog. The manuscript has been adjusted to better reflect this insight (Lines 425-429).

Round 2

Reviewer 1 Report

Comments and Suggestions for Authors

Thanks for addressing the comments so the paper is fairly balanced. 

Author Response

We would like to thank the reviewers for all of the positive feedback.